# Coming from the Wild: Multidrug Resistant Opportunistic Pathogens Presenting a Primary, Not Human-Linked, Environmental Habitat

**DOI:** 10.3390/ijms22158080

**Published:** 2021-07-28

**Authors:** Fernando Sanz-García, Teresa Gil-Gil, Pablo Laborda, Luz E. Ochoa-Sánchez, José L. Martínez, Sara Hernando-Amado

**Affiliations:** Centro Nacional de Biotecnología, CSIC, 28049 Madrid, Spain; fsanz@cnb.csic.es (F.S.-G.); tgil@cnb.csic.es (T.G.-G.); plaborda@cnb.csic.es (P.L.); luzedith.os@gmail.com (L.E.O.-S.); shernando@cnb.csic.es (S.H.-A.)

**Keywords:** opportunistic pathogens, MDR, One-Health, intrinsic resistance, environmental bacteria, *Pseudomonas aeruginosa*, *Stenotrophomonas maltophilia*, *Acinetobacter baumannii*, *Burkholderia cepacia*, *Shewanella*, *Aeromonas*

## Abstract

The use and misuse of antibiotics have made antibiotic-resistant bacteria widespread nowadays, constituting one of the most relevant challenges for human health at present. Among these bacteria, opportunistic pathogens with an environmental, non-clinical, primary habitat stand as an increasing matter of concern at hospitals. These organisms usually present low susceptibility to antibiotics currently used for therapy. They are also proficient in acquiring increased resistance levels, a situation that limits the therapeutic options for treating the infections they cause. In this article, we analyse the most predominant opportunistic pathogens with an environmental origin, focusing on the mechanisms of antibiotic resistance they present. Further, we discuss the functions, beyond antibiotic resistance, that these determinants may have in the natural ecosystems that these bacteria usually colonize. Given the capacity of these organisms for colonizing different habitats, from clinical settings to natural environments, and for infecting different hosts, from plants to humans, deciphering their population structure, their mechanisms of resistance and the role that these mechanisms may play in natural ecosystems is of relevance for understanding the dissemination of antibiotic resistance under a One-Health point of view.

## 1. Introduction

Bacterial organisms causing human infections can be divided into two categories; those that infect healthy people, and those that mainly infect people with underlying diseases, immunosuppressed or debilitated. While the former are relevant both in the community and in the hospitals, the latter have been dubbed opportunistic pathogens and are primarily a hospital problem [1]. Opportunistic pathogens have historically originated from human commensal bacteria. Indeed, in the seminal paper that led to the search of antibiotic producers in soils, the main reason for such screening was that, despite the soil being a sink for organisms infecting humans, “one hardly thinks of the soil as a source of epidemics” [2]. Nevertheless, in the last decades, an increased prevalence of opportunistic pathogens with an environmental origin, most of them non-fermentative Gram-negative bacteria [3], has been reported [4]. Most of these pathogens present low susceptibility to antibiotics currently used in therapy, suggesting that the enrichment of these pathogens at hospitals can result from the selection pressure exerted by antibiotics used for treating infectious diseases [5]. Actually, one of the risk factors for being infected by these pathogens is previous antibiotic treatment with broad-spectrum antibiotics. As opposed to what Waksman and Woodruff stated in 1940 [2], it is now evident that natural environments encompass an undefined reservoir of bacterial species, some of which have the potential to infect humans. These infections mainly occur in immunodeficient people and patients with underlying diseases. This fact suggests that, beyond the existence of specific lineages that have evolved towards virulence, the main reason behind infection by this type of opportunistic pathogens is the health status of the infected patient. Indeed, for most of the opportunistic pathogens herein reviewed, there are not clear genomic differences between environmental and clinical isolates. This factor does not mean that epidemic clones are absent, but rather that those clones, more frequently involved in outbreaks at hospitals, are also present in natural ecosystems. Further, the fact that most of their virulence determinants and several antibiotic resistance genes (ARGs) are usually present in their core genomes supports that these elements have evolved to deal with functions other than infecting humans in the natural habitats that these microorganisms colonize. In the present article, we review the most relevant current information on these pathogens, with a particular emphasis on their mechanisms of antibiotic resistance (AR). It is important to notice that, besides being the primary habitat (i.e., the origin) of some opportunistic pathogens, natural ecosystems are the places where all human bacteria, pathogens and commensals end up, along with the ARGs they carry [6,7,8]. While environmental antibiotic-resistant organisms, such as *Pseudomonas aeruginosa* or *Burkholderia cepacia,* regularly colonize environmental habitats, other pathogens with relevance for the dissemination of resistance, such as *Escherichia coli*, *Enterococcus* or *Klebsiella pneumoniae,* are part of human-linked microbiomes; their finding in a natural ecosystem is considered a sign of anthropogenic pollution [9,10], to the extent that it has been stated that resistant organisms detected in wastewater treatment plants should reflect the overall resistome of the human populations they serve [11,12,13,14]. Certainly, upon such pollution, natural ecosystems can be drivers for the evolution and spread of AR in any human pathogen [6,15]; however, in the current review, we focus just on those organisms that present a bona fide, non-clinical, environmental primary habitat, where they have evolved [16,17] before causing human infections. 

## 2. *Pseudomonas aeruginosa*

*P. aeruginosa* is a Gram-negative, rod-shaped, non-fermentative, facultative anaerobic bacterium able to colonize a wide range of different habitats due to its high metabolic versatility and broad capacity of adaptation to fluctuating environments [18]. Its presence has been described in soil [19], crude oil [20] or different aquatic systems such as wastewater [21], freshwater and seawater environments [22], being found among the most frequent locations those closely related to human activities [23].

Considering a host as an environment colonizable by bacteria [16], the ability of *P. aeruginosa* to cause infections in a large range of hosts may also be used to exemplify its high adaptability and ubiquitous distribution. *P. aeruginosa* infections have been reported in plants [24], animals -such as insects [25], nematodes [26], fishes [27] or mammals [28], including humans [29]- or even amoebas, such as *Dictyostelium discoideum* [30]. However, despite the wide distribution of *P. aeruginosa*, several studies have indicated that there are no specific clones associated with specific habitats and that environmental and clinical isolates are indistinguishable. In addition, there is a consensus about the non-clonal epidemic nature of *P. aeruginosa* population structure [31]. 

*P. aeruginosa* is one of the main causes of nosocomial infections, including acute respiratory diseases and bacteraemia [32]. Moreover, it can chronically infect immunocompromised people or patients with underlying diseases, such as chronic obstructive pulmonary disease (COPD) [33], cystic fibrosis (CF) [34], AIDS [35], cancer [36] or those presenting burn or surgical wounds [37], being that these infections are an utmost source of morbidity and mortality in intensive care units (ICUs). 

The impact of *P. aeruginosa* on human health cannot be understood without taking into account the vast amount of virulence factors it possesses. Proteases, flagella, secretion systems, biofilm formation (which is particularly worrying when located in catheters, prosthesis or lungs [38,39,40]) or quorum sensing (QS) (the cell-cell signalling system that coordinates the expression of most of the said factors [41]), are elements that pave the way for infection and hamper therapies. Further, this microorganism exhibits low susceptibility to a great number of drugs [42], an issue that is dissected below. Overall, these features result in *P. aeruginosa* being subsumed into two bacterial ensembles, namely ESKAPE (acronym of *Enterococcus faecium*, *Staphylococcus aureus*, *K. pneumoniae*, *Acinetobacter baumannii*, *P. aeruginosa*, and *Enterobacter* spp.) and TOTEM (TOp TEn resistant Microorganisms), which include the currently most relevant multidrug-resistant human pathogens [43,44].

*P. aeruginosa* infections are frequently treated with aminoglycosides, especially tobramycin, as well as with cephalosporines or β-lactam/β-lactamase-inhibitor combinations, such as piperacillin/tazobactam or ceftazidime/avibactam [32]. Besides, fluoroquinolones (ciprofloxacin), polymyxins, fosfomycin, aztreonam and carbapenems are also antibiotics of choice, which usage depends on the characteristics of the infection [45]. Nevertheless, the aforementioned intrinsic resistance of this pathogen to some antibiotics has compelled many to search for novel β-lactam/β-lactamase inhibitors, like imipenem/relebactam [46], or the development of new antimicrobial compounds, such as plazomicin, murepavadin or doripenem [47]. It is also important to remark that non-antibiotic therapies have been delved into, in order to counteract treatment failure when resistance to classical drugs emerges. Among these strategies, anti-virulence compounds, efflux pump inhibitors and permeabilizing membrane compounds (co-administered with antibiotics or on their own) stand out as the most promising alternatives [48,49,50]. In addition, evolution-based approaches that exploit phenotypic convergence and negative hysteresis phenomena are currently being investigated to fight *P. aeruginosa* and other human pathogen infections [51,52,53,54].

The above-stated high intrinsic resistance to antibiotics of *P. aeruginosa* (Figure 1) is due to its low outer membrane permeability [55], the production of antibiotic-modifying enzymes [56], and the large stock of multidrug resistance (MDR) efflux pumps it harbours [47]. Concerning the latter, there are 12 Resistance Nodulation Division (RND) family members that have been ascribed to this bacterium; among them, MexAB-OprM, MexXY-OprM, MexCD-OprJ and MexEF-OprN are of significant interest, given their known role in clinical settings [57]. The first two are the ones that have been shown to contribute to intrinsic resistance, but every system is able to extrude a wide range of antimicrobial agents (Table 1). Regarding antibiotic-modifying enzymes, *P. aeruginosa* can resort to its inherent assortment of β-lactamases, making AmpC the most noteworthy [58], and aminoglycoside-modifying enzymes, namely aminoglycoside acetyltransferases, phosphotransferases and nucleotidyltransferases [59]. Additionally, it is worth emphasizing that its intrinsic resistome does not only consist of classical resistance determinants, but it may encompass basic components of bacterial physiology [60]. For example, Crc is a global regulator of carbon metabolism whose inactivation entails an increased susceptibility to several antimicrobials in *P. aeruginosa* [61]. This situation agrees with the notion that the ancestral, physiological function of intrinsic resistance determinants of pathogens with an environmental origin goes beyond counteracting the activity of antimicrobial agents currently used in therapy. As previously discussed [42], interfering with the effectiveness of antibiotics is a novel functional role of these determinants, promoted by the current antibiotic era.

In addition to intrinsic AR, increased resistance levels may also be acquired by chromosomal mutations that boost the expression of the above-described determinants (Figure 1), a situation that frequently takes place during chronic infections [62]. For instance, mutations in genes encoding regulators of MDR efflux pumps can lead to the overexpression of the latter and, as a consequence, a more efficient extrusion of drugs. In the case of *mexAB-oprM,* overexpression may be due to mutations in the gene that encodes its local repressor, MexR, or in genes encoding other secondary regulators, NalC or NalD, events that have been reported in vivo [63,64]. Similarly, *mexCD-oprJ* and *mexXY-oprM* overexpression could be driven by spontaneous mutations in genes encoding their repressors, NfxB and MexZ, respectively [63,65]. Concerning β-lactamases, indirect or direct repressors of *ampC* expression, AmpR or AmpD, are commonly found mutated in β-lactam resistant strains of *P. aeruginosa*, presenting an enhanced β-lactamase activity and, consequently, resistance to β-lactams [58]. 

Besides, mutations can be selected in genes that encode the resistance determinants themselves. In the case of MDR efflux pumps, it has been published that amino acid changes in MexY, a subunit of the MexXY-OprM pump, may optimize antibiotic recognition site and, hence, improve drug efflux [66]. Regarding enzymes, cephalosporinase AmpC variants are able to extend their substrate spectrum, becoming capable of hydrolyzing carbapenems [67] or recent β-lactam/β-lactamase inhibitor combinations [68]. 

Beyond modifications in resistance determinants, the mutational resistome of *P. aeruginosa* is still more multifarious. Some examples of the versatile mutational resistome of this opportunistic pathogen are the mutations in genes involved in the peptidoglycan recycling pathway, as *mpl or dacB*, which raise β-lactamase activity [69]; in genes that encode drug targets, as *gyrA* or *gyrB*, which foster quinolone resistance [70]; in genes encoding Penicillin Binding Proteins (PBPs), i.e., PBP3, which cause β-lactams resistance [71]; or even in *loci* that do not seem to be associated with AR, as *pilQ*, a gene that codes for a Type IV pili protein, which can give rise to resistance against various antipseudomonal agents [72].

Alternatively, this microorganism can acquire ARGs through Horizontal Gene Transfer (HGT) [47]. These genes can locate in integrative and conjugative elements (ICEs), plasmids, integrons, transposons or prophages (Figure 1), and they can be transferred by different mechanisms [73]. As expected, there is plethora of examples of ARG acquisition by *P. aeruginosa* through HGT, with ICEs and plasmids being the most usual ARGs carriers [74]. In this sense, an extensive miscellany of β-lactamases, aminoglycoside and fluoroquinolone-modifying enzymes have routinely been detected in these vectors [75,76,77,78], either alone or accompanied by large arrays of ARGs [79]. Correspondingly, integrons, which do not transfer independently, but are gene-recruiting elements, may also harbour metallo β-lactamases (MBLs) (i.e., carbapenemases) [80], aminoglycoside-modifying enzymes [81], or both [82], among other resistance mechanisms [83]. Phage particles containing ARGs have been found in the lungs of CF patients suffering chronic infections by *P. aeruginosa* [84]; in addition, a composite phage-like plasmid carrying the β-lactamase-encoding gene *blaKPC-2* has been found in a carbapenem-resistant *P. aeruginosa* isolate [85]. However, a deep understanding of the relevance of prophages in disseminating ARGs in *P. aeruginosa* requires further studies. In conclusion, this pathogen wields an astonishing range of alternatives to achieve AR.

Lastly, from a One-Health perspective, it must be noted that *P. aeruginosa* ubiquity in nature aggravates the problem of AR. This bacterium has a broad sub-lethal selective window to different antibiotics, under which resistant mutants may arise [86,87]. This matter becomes more alarming since substantial concentrations of drugs (i.e., 31 or 61 mg/L of ciprofloxacin or tetracycline, respectively) have been detected in habitats that *P. aeruginosa* can colonize [23,88], besides clinical settings. Further, aquatic ecosystems have been suggested as reservoirs and sources of ARGs, usually carried on plasmids, a situation that may play a critical role in the propagation of antimicrobial resistance among *P. aeruginosa* strains [89].

Besides intrinsic and genetically acquired, stable AR, the resistance phenotype can be acquired transiently, without the need for genetic changes. Transient resistance can be achieved by the bulk of the population, as happens in the case of biofilms [90]. The capacity of *P. aeruginosa* to form a biofilm within a host–which contributes to its ability to inhabit diverse ecological niches- impedes phagocytosis and diminishes the efficiency of antimicrobial treatments, sometimes provoking chronic and persistent infections in host tissues or prosthetic devices [39]. Transient resistance can also be developed just by a bacterial subpopulation, a situation dubbed persistence, which is defined as the ability of a part of the bacterial population to survive under an antimicrobial treatment without acquiring genetic changes conferring resistance [91]. In *P. aeruginosa*, persistence is encountered under nutrient limiting conditions [92], as is the presence of QS signaling molecules [93]. 

One explanation for the increased antimicrobial resistance of biofilm-growing *P. aeruginosa* is the presence of cells with a slow-growing metabolic state in some parts of the biofilm, which constitute a subpopulation of persisters [94]. Other reasons for an increased transient resistance are a more difficult diffusion of compounds due to the complex structure of the biofilm [90] or the presence of elements that reduce the activity of antimicrobials within the biofilm, such as glycerophosphorylated β-(1, 3)-glucans or cyclic-β-(1, 3)-glucans, which sequester aminoglycoside antibiotics [95]. In addition, changes in the expression levels of ARGs as *mexAB-oprM* and *mexCD-oprJ* during the biofilms state of growth are also of relevance [96,97]. 

Besides persistence and the formation of recalcitrant biofilm structures, changes in the expression of AR determinants due to specific signals and/or conditions are of relevance for developing transient resistance. Indeed, this scenario may be encountered during infection, thus compromising the efficacy of antipseudomonal treatments [98]. For instance, the presence of the inducible β-lactamase AmpC in *P. aeruginosa*, the expression of which is enhanced by some β-lactam antibiotics, may lead to treatment failure due to transient β-lactam resistance [99]. Changes in the permeability of *P. aeruginosa* associated with magnesium limiting conditions transitorily reduce the negative charge of the cell surface through an up-regulation of an LPS modification operon, driving to enhanced resistance in positively charged antimicrobials, like cationic antimicrobial peptides or polymyxin B [100,101].

The role of MDR efflux pumps regarding *P. aeruginosa* transient AR should also be highlighted. Since these systems are involved in different key processes for bacterial physiology, tight regulatory control over their expression, dependent on environmental conditions, may be expected. Therefore, in some situations or in the presence of specific effectors, a temporary rise of the expression of efflux pump encoding genes is achieved [102]. Some of these inducing conditions may be found in clinical settings, allowing bacteria to resist an antimicrobial treatment through a transitory improved antibiotic extrusion capacity. For instance, expression of MexCD-OprJ efflux pump’s encoding genes is induced by molecules that *P. aeruginosa* may run into during an infection. Some of these molecules are disinfectants or anaesthetic agents (e.g., procaine or atropine), which induce quinolone resistance [103], as well as the human host defence peptide LL-37, which increases resistance towards quinolones and aminoglycosides [104].

Further, *mexAB-oprM* expression is induced under oxidative stress conditions [105] and by triclosan or pentachlorophenol [106]. Furthermore, nitrosative stress, chloramphenicol presence and contact with human airway epithelial cells are circumstances that trigger *mexEF-oprN* overexpression [107,108], supporting the idea that the MexEF-OprN efflux pump might contribute to *P. aeruginosa* transient resistance during lung infection. Another *P. aeruginosa*’s efflux pump, MexXY, may also contribute to transient AR in clinical settings since the expression of its encoding genes is induced under oxidative stress conditions or in the presence of antibiotics able to inhibit protein synthesis as aminoglycosides or tetracyclines [109].

In summary, transient resistance must not be neglected during *P. aeruginosa* infections. Different conditions, compounds and modes of growth that may take place during the infection process might transiently increase the resistance of this opportunistic pathogen to several antimicrobial treatments, thus hindering the eradication of the infecting bacterial population. Further, recent work has shown that the early appearance of tolerance mutations facilitates the evolution of AR [110], a feature of particular relevance in the case of chronic *P. aeruginosa* infections [111].

Infections by other *Pseudomonas* species with an environmental origin and biotechnological potential, as *Pseudomonas putida* [112,113,114,115], have also been reported, although their prevalence is much lower. Besides intrinsic resistance determinants [116,117], the acquisition of carbapenemases, as KPC-2 [113], constitutes an additional risk for the efficient treatment of infections by these pathogens. 

## 3. *Acinetobacter baumannii*

*Acinetobacter* is another non-fermentative Gram-negative bacterial genus, firstly reported as a significant nosocomial pathogen in the late 1970s. This microorganism harbours an entire repertoire of intrinsic resistance determinants, and it easily acquired novel ARGs soon after its detection as a cause of infections [118], becoming nowadays one of the most prevalent resistant pathogens causing problems at hospitals. Unlike other pathogens discussed in the current review, and despite the *Acinetobacter* genus being ubiquitous, the potential primary environmental niches of *Acinetobacter baumannii*, the species causing most problems at hospitals, are still not well established [119]. Hence, more studies are still needed to delimitate the outside-hospitals reservoirs of *A. baumannii* [120]. Notably, it seems that *A. baumannii* presents the largest pangenome and biochemical versatility within the species of the *Acinetobacter* genus [121,122]. Its open pangenome contains a variety of mobile genetic elements, most notably integrons, transposons and plasmids [123], which may support the capacity of this opportunistic pathogen for acquiring ARGs. Integrons and transposons can be located in genomic islands, some of which have been dubbed islands of resistance due to the presence of multiple ARGs inside them. It has been reported that around 40% of *A. baumannii* pangenome is specific to each strain [121], indicating that gene exchange within this bacterial species has a certain degree of clonal specificity. 

With regard to the core genome of *Acinetobacter*, it has been described to contain 950 families of orthologous proteins, including a large number of virulence factors [124] and at least 1590 orthologous proteins that correspond to 44% of the size of the smallest proteome of the species [121].

Due to the importance of hospital outbreaks, the population structure of *A. baumannii* is now well-established [125]. At least six major international clonal lineages (ICL), distributed across continents worldwide, have been described [125]. Three successful clones re-named as “international clones I-III”, among which ICLI and ICLII display MDR phenotypes that may be favouring their clonal expansion [126], are included in these lineages. A recent study, based on the analysis of almost 2500 genomes, shows that *A. baumannii* can be divided into two clusters. Notably, the strains of one cluster, which contain a CRISPR/Cas system, rarely harbour plasmids, indicating that CRISPR/Cas elements may modulate the acquisition of novel genes in *A. baumannii* [127].

Concerning human health, *A. baumannii* strains have been isolated primarily from hospitalized patients, and this pathogen is associated with infections of the respiratory tract, bloodstream, wound, skin and soft tissue, urinary tract and central nervous system [128]. Besides humans, *A. baumannii* has been isolated in veterinary medicine, infecting seriously ill animals [129], livestock and wildlife; thereby indicating that this opportunistic pathogen constitutes a One-Health problem [130]. 

The main limitation regarding the treatment of *A. baumannii* infections is the increasing prevalence of MDR isolates. Intrinsic determinants, such as the OXA-type *Acinetobacter*-derived cephalosporinase [131] or the RND efflux pumps AdeABC [132], AdeFGH and AdeIJK [133], stand out as major determinants of intrinsic AR in this bacterium. While the first efflux pump contributes to acquired resistance when overexpressed, the contribution of the second to this phenotype is less relevant because AdeIJK overexpression is toxic above a given threshold. AdeFGH also confers MDR when overexpressed, while some non-RND efflux systems, such as CraA, AmvA, AbeM and AbeS, have been described to be involved in *A. baumannii* AR too [134]. 

Notably, it has been shown that one-step AR mutations can be selected in vivo during the treatment of the infected patients [135]. Among them, mutations in the genes encoding the regulators of the expression of MDR efflux pumps lead to their overproduction and to associated cross-resistance to a variety of antimicrobials [136]. In addition, mutations in genes encoding outer membrane proteins, such as OmpA, CarO and OprD, also contribute to AR and modulate virulence of this opportunistic pathogen [137], providing an example of the crosstalk between virulence and AR [138].

Besides intrinsic and mutational acquired resistance, the members of this bacterial species have acquired several β-lactamases and other ARGs by HGT [139]. Among them, and in addition to the intrinsic OXA-type β-lactamase, other OXA derivatives have been found [140], frequently linked to insertion sequences (ISs) located upstream the genes encoding β-lactamases [141]. Indeed, the activity of ISs, capable of modifying the expression of genes involved in resistance when located in the right positions, seems to be also instrumental for the acquisition of the resistance phenotype [142,143]. Despite AR plasmids not being as frequent in *A. baumannii* as in *Enterobacteriaceae* [144], the plasticity of its genome [145] allows the acquisition of ARGs, many of them present in transposons and in integrons, within plasmids and the chromosome. In this regard, it is worth mentioning that more than 130 gene cassettes containing ARGs have been identified in integrons located in *A. baumannii* genomes [146]. Although several studies have shown the high prevalence of class 1 integrons, which often contain resistance gene cassettes, other studies carried out in Latin American countries, such as Chile, Argentina and Brazil, have also shown a wide distribution of class two integrons in this species [147,148]. It is important to notice that while extended-spectrum β-lactamases (ESBLs) as ESBLs PER-, GES- and VEB-type are the most common *A. baumannii*, TEM- and SHV-type ESBLs, the most prevalent in *Enterobacteriaceae*, are less frequently found in *A. baumannii*, supporting that gene exchange between these two groups of microorganisms is likely low [149], although still possible (see the example of NDM1 below). 

Notably, ARGs acquired by *A. baumannii* are frequently clustered, forming part of genomic islands, dubbed AbaRs. These AbaRs [146] present backbones resembling Tn*6019*, Tn*6022* and Tn*6172* transposons [150] and seem to be clone-specific [151]. This feature may mean that, once an AbaR has been acquired, its mobilization to another phylogenomic *A. baumannii* group could be limited. Hence, HGT via plasmids or other mobile genetic elements might be on the basis of the acquisition of resistance by *A. baumannii* [152]. In addition, recent works indicate that ARGs-containing bacteriophages might contribute to AR spread in this microorganism [153,154]. Finally, it has been recently found that this organism can be naturally competent [155,156], opening the possibility that direct transformation could be a relevant mechanism triggering the acquisition of ARGs by this bacterium.

Currently, more studies have shown the importance of other bacterial species within the *Acinetobacter* genus in clinical settings [157]. Along with *A. baumannii*, other species such as *Acinetobacter pittii* and *Acinetobacter nosocomialis* have been frequently isolated in patients [158]. *A. pittii* was isolated in China, and its potential to acquire resistance to carbapenems by a mutation in *bla*_OXA-499_ has been observed [159]. Regarding *A. nosocomialis,* the importance of the RND-type efflux pumps, AdeIJK and AdeABC, in its resistance phenotype has been highlighted [160].

Other species of the genus have also been described in natural and clinical environments, such as *Acinetobacter soli,* firstly isolated from a Korean forest [161] and identified in domestic animal lice [162]. Although the first reports indicated that the microorganism came from environmental sources, it has also been found in clinical settings [163]. In China, an MDR isolate of this bacterial species containing the β-lactamase encoding genes (*bla*_OXA-58_, *bla*_IMP-1_, *bla*_NDM-1_ and *bla*_TMB-2_) caused the death of a patient under treatment [164]. Notably, it has been suggested that *bla*_NDM-1_ is a chimaera constructed in *A. baumannii*; a feature supporting that this species can be the origin as well as a reservoir for the transfer of this relevant carbapenemase of *Enterobacteriaceae* [165]. This has been further reinforced with in vitro data indicating that *Acinetobacter* plasmids could have contributed to the spread of *bla*_NDM-1_ in *Enterobacteriaceae* [166]. Also supporting this idea is the characterization of transferable plasmids containing *bla*_NDM-1_ in both *A. soli* and *A. pittii* and their mobility between the genus [167]. *A. pittii* isolates containing plasmids belonging to new incompatibility groups, which carry genes encoding OXA-type carbapenemases, and with the ability to transfer them to other species of the genus, have been reported too [168]. 

The possibility that plasmids carried by *Acinetobacter* spp. might be transferred to other human pathogens, hence contributing to ARGs spread, has been analysed in other studies. For instance, genomic analyses comparing *Acinetobacter* spp. clinical and environmental (water and soil) isolates suggested that Rep_3-type plasmids can be transferred between *Acinetobacter* spp. and bacteria belonging to other genera from different environments [169]. Further, the in silico analysis of 173 plasmids of a wide variety of sizes from 17 countries showed that some plasmid lineages have the capacity to replicate in many bacterial genera, while others only do it within species of the *Acinetobacter* genus [170]. It is important to notice that the number of different plasmid lineages harboured by *A. baumannii* is low, with around one-third of them containing ARGs and that gene flux among different plasmids seems to be mediated by transposons [170].

Reports on *Acinetobacter* spp. carrying ARGs continue to increase. Examples of them are some strains of *Acinetobacter bereziniae*, recently isolated from human clinical samples [171], carrying *bla*_OXA-type_ [172] and MBLs encoding genes, as *bla*_NDM-1_; in both cases encoded in plasmids [173]. Another example is *Acinetobacter junii* strains, presenting *bla*_NDM-1_ and *bla*_OXA-58_, and isolated from hospitals [174].

## 4. *Stenotrophomonas maltophilia*

*Stenotrophomonas maltophilia* is a ubiquitous non-fermentative Gram-negative microorganism described in a variety of environments [175], from natural to anthropogenic niches, such as soil [176], water [177] or sediments [178,179]. Besides its role as an opportunistic pathogen, *S. maltophilia* is also a plant endophyte, and different strains with biotechnological value have been described [180]. This feature makes particularly relevant to distinguish between infective and non-infective *S. maltophilia* strains. However, there is evidence of epidemic *S. maltophilia* lineages [181] and it seems that, as has been described for *P. aeruginosa* [182,183], there are not specific clades evolving towards virulence [184,185]. Instead, the prevalence of *S. maltophilia* infections mainly derives from the underlying health condition of the patient, more than from specific characteristics of the isolate causing such infection. Through different genotyping methodologies, it has been described that the *S. maltophilia species complex* (Smc) contains multiple genospecies [178,186,187]. Four genospecies belong to the *S. maltophilia sensu*
*stricto* species, which is the main cause of infections in humans and the only one that consistently expresses MBLs (Sgn1, Sgn2, Sgn3 and Sgn4) [178]. The Smc displays high genetic, ecological and phenotypic diversity [188,189] as well as heterogeneous resistance and virulence phenotypes [187,190]. In fact, this phenotypic heterogeneity mainly results from problems in species delimitation within the Smc [178,191]. This problem can be aggravated since, based on estimates of genomic mean nucleotide identity values >94%, it has been recently proposed to reclassify *Stenotrophomonas africana*, *Pseudomonas beteli* and *Pseudomonas hibiscicola* as *S. maltophilia* [188,192]; despite these species are not known to be a relevant cause of human infections.

It was in the 1980s when *S. maltophilia* became significantly reported as an emerging pathogen. Nowadays, it has become the third most common cause of nosocomial infections caused by non-fermentative Gram-negative bacilli. Even though it is not considered a highly virulent bacterium, it is one of the leading drug-resistant pathogens of more significant public health concern in hospitals worldwide and is associated with mortality rates between 14 and 69% in patients with bacteraemia. Although it is mainly a nosocomial pathogen, community-acquired infections are an increasing trend [193]. *S. maltophilia* is mostly associated with respiratory infections and acute exacerbations of COPD, followed by bloodstream infections. Less frequently, it causes infections of the skin and soft tissues, biliary and urinary tract, endocarditis, meningitis, intra-abdominal infections and endophthalmitis [194]. The most affected patients are those with previous pathologies (CF, HIV infection or cancer–particularly obstructive lung cancer-), mechanical ventilation, indwelling catheters, corticosteroid or immunosuppressant therapy, together with those hospitalized for prolonged periods or ICU admission and previous broad-spectrum antibiotics therapy [5]. In these patients, *S. maltophilia* infections are associated with high mortality rates [5,175,195]. Infections caused by this microorganism occur in adults and children, and the transmission to susceptible individuals takes place through direct contact with the source. Possible sources are hands of health care professionals, aerosols from CF patients, suction system tubing of dental chair units, contaminated endoscopes or tap water [175].

Treatment of infections caused by *S. maltophilia* is complicated given the intrinsic resistance mechanisms against most antimicrobials that this bacterium presents [196,197]. Trimethoprim/sulfamethoxazole (SXT) is currently the treatment of choice [198]; albeit, combination therapies of SXT plus ciprofloxacin, ceftazidime, tobramycin or tigecycline, which exhibit a greater activity than SXT alone, are also implemented [199,200]. However, the acquisition of resistance to SXT limits its use. Therefore, new therapeutic options are needed to tackle these infections. Ticarcillin/clavulanate or ceftazidime in combination with ciprofloxacin are the agents used in most SXT resistant infections. On the one side, ceftazidime and ticarcillin/clavulanate used to be the most effective β-lactams against *S. maltophilia*, but the number of resistant isolates is increasing [5]. On the other side, ciprofloxacin or newer fluoroquinolones as levofloxacin are still a helpful alternative, even though the number of resistant isolates is sizable [201]. Ultimately, recent studies have shown that minocycline [202] and colistin, alone or in combination with N-acetylcysteine [203], could be used for treating infections caused by this microorganism. 

Genome sequencing of *S. maltophilia* clinical [195] and environmental isolates [204] indicated that several of the elements involved in the characteristic AR phenotype of this bacterial species are shared by strains isolated from different habitats. Therefore, these elements have evolved before the use of antibiotics for human therapy, as described for *P. aeruginosa*. In all these genomes, many genes encoding determinants of resistance to antibiotics (Table 2), such as β-lactams, cephalosporins, macrolides, fluoroquinolones, aminoglycosides or carbapenems, have been found [195,205,206]. These data show that *S. maltophilia* intrinsic resistance has not been acquired upon evolution in the presence of antibiotics currently used in therapy, although increased levels of resistance can be acquired by mutations or by ARGs acquisition through HGT [194,206,207]. 

Concerning intrinsic resistance, *S. maltophilia* possesses two inducible β-lactamases, L1 and L2. L1 is a broad spectrum (excluding monobactams) MBL [208], while L2 is classified as a class A clavulanic acid-sensitive cephalosporinase [209]. The expression of these enzymes, mostly controlled by the AmpR transcriptional regulator, is directly induced by the antibiotics they provide resistance to [210,211]. AmpR acts as an activator in the presence of inducers, such as β-lactams, but in the absence of them, it is a repressor of *L2* expression [212]. 

In addition, aminoglycoside-modifying enzymes encoded in the *S. maltophilia* genome confer low susceptibility to several aminoglycosides [213]. Three of these enzymes have been analysed: N-aminoglycoside acetyltransferases AAC(6′)-Iz that contributes to resistance to amikacin, tobramycin, sisomicin and netilmicin [198,214]; AAC(6′)-Iak [198,213] that decreases susceptibility to several aminoglycosides, including arbekacin, kanamycin, neomycin, sisomicin or tobramycin [213]; and the aminoglycoside phosphotransferase APH(3′)-IIc [215], that confers resistance to kanamycin, neomycin, paromycin and butirosin [215].

Another mechanism of intrinsic resistance in this bacterium is the chromosomally-encoded SmQnr protein, which contributes to intrinsic resistance to quinolones [216,217,218] by protecting DNA gyrase and topoisomerases from fluoroquinolones’ activity [219].

Along with the inactivating enzymes, the major contributors to intrinsic resistance to many antimicrobial agents in *S. maltophilia* are the chromosome-encoded MDR efflux pumps. The best-characterized group of pumps is the RND family. Eight of these complexes (SmeABC, SmeDEF, SmeGH, SmeIJK, SmeMN, SmeOP, SmeVWX and SmeYZ) are encoded in the genome of *S. maltophilia*, and the role in AR of seven of them has been studied [220]. Only when the expression level is constitutively significant, as it happens with SmeYZ, SmeDEF, SmeGH, SmeIJK and SmeOP [220,221,222,223,224], do these pumps contribute to intrinsic resistance. The SmeYZ system is involved in intrinsic resistance to aminoglycosides, tetracycline, leucomycin and SXT [220]. SmeDEF overexpression is linked to quinolones, chloramphenicol, tetracycline, tigecycline, macrolides, sulfamethoxazole, trimethoprim and SXT resistance [207,225,226,227], as well as to resistance to the biocide triclosan [228,229]. SmeGH is involved in intrinsic resistance to β-lactams, quinolones, tetracycline and polymyxin B, as well as to other toxic compounds, such as menadione, tert-butyl hydroperoxide, naringenin and hexachlorophene [221]. For their part, SmeIJK and SmeOP confer resistance to aminoglycosides, tetracycline, ciprofloxacin, levofloxacin, leucomycin or minocycline [222] and nalidixic acid, doxycycline, aminoglycosides or macrolides [223], respectively.

Along with this well-characterized group, other MDR efflux pumps are encoded in the *S. maltophilia* genome. Among them, ATP binding cassette (ABC) efflux pumps such as, SmrA which contributes to fluoroquinolones, tetracycline and doxorubicin resistance [230]; and MacABCsm, which is involved in aminoglycosides, macrolides and polymyxins resistance [231]; and the major facilitator superfamily (MFS)-type efflux pump EmrCAB, implicated in the extrusion of nalidixic acid, erythromycin and tetrachlorosalicylanilide [232], are of clinical relevance. 

This chromosomally-encoded arsenal of resistance elements, together with their low-permeability membranes, are responsible for *S. maltophilia*’s MDR intrinsic phenotype that is independent of the environment in which this bacterium lives [233]. 

In addition to their contribution to intrinsic resistance, these elements also contribute to acquired resistance when overexpressed or mutated [234]. The overexpression of either β-lactamases or MDR efflux pumps stands as the main cause of the acquisition of resistance in clinical isolates of *S. maltophilia*. In the case of efflux pumps, increased AR is associated with their overexpression, mostly by the acquisition of mutations in their regulators. The selection of these mutations, leading to efflux pumps’ overexpression, is particularly problematic since they lead to an MDR phenotype. For example, the most prevalent cause of *S. maltophilia* acquired resistance to quinolones is the overproduction of SmeDEF, mainly by mutations in the gene encoding the negative regulator SmeT [224], and of SmeVWX, by mutations in the gene encoding its SmeRv regulator [225]. Importantly, *S. maltophilia* is the only known bacteria in which high-level resistance to quinolones is only due to the overexpression of MDR efflux pumps, not to mutations in genes encoding quinolones targets [235,236]. Besides, evolution experiments made in the presence of tigecycline have revealed that mutations in *smeT,* leading to SmeDEF overexpression, constitute the first step in *S. maltophilia* tigecycline acquired resistance. In addition, amino acid substitutions in the gene encoding the efflux pump and, thus, changes in its structural elements are also on the basis of acquired resistance. For instance, *smeH* mutations are involved in the acquisition of resistance against ceftazidime, leading to cross-resistance towards other antibiotics, mainly β-lactams [237]. SmeABC, SmeIJK and SmeYZ also contribute to acquired resistance towards aminoglycosides, β-lactams and fluoroquinolones [238] when overexpressed.

Mutations in the antibiotics’ target genes are another cause of acquired resistance in this bacterium. Apart from the aforementioned *smeT* mutations, ribosome 30S mutations, the target of tigecycline, are among the mechanisms of tigecycline acquired resistance. Additionally, the inactivation of central carbon metabolism enzymes has also been shown to be responsible for the acquisition of AR by *S. maltophilia.* A group of in vitro selected mutants in which genes encoding the enzymes Eno, GmpA, GapA and Pgk were inactivated has allowed the study of mutation-driven fosfomycin resistance. This study showed that the inactivation of the Embden-Meyerhof-Parnas metabolic pathway is on the basis of this resistance [239].

Finally, not only mutations but also HGT contributes to *S. maltophilia* acquired resistance. Resistance to SXT may occur by the acquisition of the *sul* and *dfrA* genes present in integrons or plasmids [225]. Besides, mobile elements involved in SXT resistance, plasmid-mediated quinolone resistance genes (e.g., *qnrS* [240]), and β-lactamases (e.g., *bla_CTX-M-Gp1_* [241]), have been found in *S. maltophilia* isolates. Despite the fact that a lysogenic phage containing the dihydrofolate reductase encoding the *folA* gene has been described to contribute to trimethoprim resistance in an *S. maltophilia* isolate [242], the role of these elements in the dissemination of ARGs in this microorganism remains to be studied in detail.

As in other species, different factors, such as medium composition, osmolarity or ionic concentrations, can induce *S. maltophilia* transient AR. Temperature can also modify its antibiotic susceptibility by alterations in the outer membrane LPS conformation. For instance, this bacterium is more susceptible to aminoglycosides at 37 °C than at 30 °C since the binding and/or uptake of the antibiotic is inhibited at a lower temperature [243]. The ability to form biofilms, which reduces antibiotics’ susceptibility, is a significant feature of *S. maltophilia*. Environmental factors, such as phosphate or chloride concentrations, temperature and aerobic or anaerobic conditions, can influence the production of biofilms, being enhanced under aerobic conditions [175]. *S. maltophilia* and *P. aeruginosa* can grow together inside dense polymicrobial biofilms in different environments, including the lungs of CF patients. This kind of growth influences their behaviour, including antibiotic susceptibility [244]. Inside these biofilms, *S. maltophilia* produces a diffusible signal factor that *P. aeruginosa* senses through the two-component sensor BptS, leading to the increased production of proteins implicated in polymyxin and colistin resistance [245]. 

β-lactamases and MDR efflux pumps also contribute to transient AR since their expression is inducible. The expression of β-lactamases is induced by β-lactams [210], and the MDR efflux pump’s expression increases due to the effect of different molecules. On the one side, *smeDEF* expression is induced by plant-derived flavonoids [246] or biocides like triclosan [229]. These molecules can bind the *smeDEF* repressor, SmeT, inducing the expression of this efflux pump and reducing *S. maltophilia* quinolone susceptibility. On the other hand, fluorescence-based analyses have uncovered *smeYZ* and *smeVWX* inducers involved in aminoglycosides and chloramphenicol or quinolone resistance, respectively. Boric acid, erythromycin, chloramphenicol and lincomycin are inducers of *smeYZ* [247], whereas vitamin K3 and its analogues vitamin K2 and plumbagin, as well as iodoacetate, clioquinol and sodium selenite, are *smeVWX* inducers [248]. Finally, the tripartite efflux pump (FuaABC), related to ABC efflux pumps, whose expression is induced by fusaric acid, contributes to transient resistance to this compound [249].

## 5. *Burkholderia cepacia* Complex

The *B. cepacia* complex (Bcc) is a group of closely related non-fermenting Gram-negative bacilli that comprises 22 validated species. The taxonomy of these bacteria is complex and continuously changing [250]. This complex is formed by nine genomovars, namely *B. cepacia* (formerly genomovar I), *Burkholderia multivorans* (II), *Burkholderia cenocepacia* (III), *Burkholderia stabilis* (IV), *Burkholderia vietnamiensis* (V), *Burkholderia dolosa* (VI), *Burkholderia ambifaria* (VII), *Burkholderia anthina* (VIII), *Burkholderia pyrrocinia* (IX) and the group or taxon K (recently split into two species: *Burkholderia contaminans* and *Burkholderia lata*) [250,251,252,253,254]. 

This complex has a versatile metabolism that allows it to colonize a great variety of niches [251,252,253,254]. Moreover, the complex includes species that are important opportunistic human pathogens of CF patients [252,255] or chronic granulomatosis disease, and critical nosocomial pathogens causing bacteraemia or urinary tract infections [256,257]. Although it has been stated that infections by *B. cepacia* could be associated with a fast lung decline and increased mortality of CF patients, dubbed the cepacia syndrome [258], recent works suggest that this statement might not always be true [259]. Further, since Bcc prevalence increases with age, lung deterioration and lung transplantation [260], these underlying conditions might also be contributing to the bad prognosis of Bcc infected patients. Although *B. cenocepacia* has been traditionally the most predominant cause of infections, *B. multivorans* is increasingly being recovered from the lungs of CF patients [261]. In addition, other members of the Bcc complex, as *B. contaminans,* cause infections in CF patients too [262,263]. Besides, outbreaks of healthcare-associated Bcc infections due to the contamination of pharmaceutical products have also been reported [264,265]. 

In addition to their relevance for human health, species of this complex are also important in agriculture because of their biocontrol and biotechnological properties. Genomovar III has been identified as a commensal of different soil types and the rhizosphere of several cultivated plants, such as maize, wheat and lupin, in natural environments [266,267]. Multilocus sequence typing (MLST) analysis of environmental and clinical isolates showed that at least 20% of the strains causing human infections are also found in nature [268]. Further, soil isolates can produce infections in both plants and animals [269]. Altogether, these findings indicate that natural ecosystems constitute a reservoir of Bcc strains with clinical relevance.

Due to its versatility and relevance in clinic and natural environments and the potential biotechnological application of some strains, the population structure of *B. cenocepacia* (genomovar III) has been studied in detail. It has been found that the population is in linkage disequilibrium and presents a clonal structure, with three major clones displaying variable degrees of recombination distributed worldwide [270]. 

One of the problems associated with Bcc infections is the low susceptibility to several antibiotics (e.g., carboxypenicillins, first and second generation cephalosporins, tetracycline or tobramycin) that this bacterial group possesses. Particularly relevant is the intrinsic resistance they all have to the last resort antibiotics polymyxins [271]. The main cause of the lack of activity of this drug against Bcc relies on the particular LPS structure of this group of microorganisms. It has been shown that the addition of 4-amino-4-deoxy-l-arabinose (Ara4N) to the lipid A component of the LPS reduces polymyxin susceptibility in different organisms [272,273]. While Ara4N synthesis is usually dispensable in different bacteria, the Ara4N biosynthetic gene cluster seems to be essential for *B. cenocepacia* [274]. This fact supports that the natural incorporation of Ara4N into lipid A is likely a major cause of Bcc polymyxin intrinsic resistance. 

Like several other bacteria, Bcc presents in its genome genes encoding different AR determinants, including efflux pumps and inducible class A and class C β-lactamases, as PenB [275] (formerly dubbed PenA [276]) or AmpC [277], respectively, which expression is coregulated [278]. Recent work suggests that β-lactam inhibitors such as relebactam, enmetazobactam, avibactam or vaborbactam, might be useful for increasing the susceptibility to β-lactams of Bcc isolates [279]. 

Beyond β-lactamases, several efflux pumps are encoded in Bcc genomes. Among these systems, NorM, a member of the multidrug and toxic compound extrusion (MATE) family, has shown to play a role in polymyxin resistance, together with the aforementioned Lipid A modification [280]. Within the identified efflux pumps encoded in Bcc genomes, those present in *B. cenocepacia* stand out as the best studied. Namely, fourteen RND efflux pump-encoding genes have been found in the genome of *B. cenocepacia* [281]. These elements are able to confer resistance to clinically relevant antibiotics as aminoglycosides, chloramphenicol, fluoroquinolones and tetracyclines [281,282]. Among them, at least three are involved in intrinsic AR [283]. Notably, an MFS immunodominant efflux pump, named BcrA and involved in tetracycline and quinolones resistance, has been detected in CF patients infected with Bcc [284], suggesting that BcrA may have a relevant role in Bcc resistance to antibiotics in said patients. Besides, the finding that salicylate, a siderophore produced by *B. cenocepacia,* may induce the expression of an antibiotic efflux pump that confers resistance to chloramphenicol, trimethoprim and ciprofloxacin, suggests that this MDR element can be overexpressed; hence contributing to Bcc transient resistance in environments with low iron availability, such as CF patients’ lungs [285].

Besides classical AR determinants, the low susceptibility to antibiotics of Bcc is also due to global mechanisms of response to stressful compounds, such as the production of lipocalins, a family of small proteins capable of binding hydrophobic ligands. It has been shown that a soluble *B. cepacia* lipocalin, produced in the presence of antibiotics, allows the sequestration of such antibiotics, hence contributing to Bcc resistance [286].

Regarding genetic changes leading to acquired resistance, mutation and recombination stand as major players in this process. It has been found that *B*. *multivorans* diversifies into various clones presenting different phenotypes when causing chronic infections in CF patients; and that some *loci* involved in β-lactams resistance present multiple mutations in recombinogenic regions [287]. Among those, mutations in *ampD,* which encode a transcription factor that coregulates the expression of the two intrinsic β-lactamases AmpC and PenB, stand out among the major causes of resistance to β-lactams in Bcc. Notably, *ampD* is highly prone to acquire AR mutations with an estimated frequency in the range of 10^−6^ to 10^−5^ [278].

In addition to mutation-driven AR, the acquisition of ARGs through HGT by some Bcc strains has been reported. Amongst them, Type I integrons, containing the sulphonamide resistance gene *sul1* and carrying the aminoglycoside resistance genes *aacA4* or *aacA7*, or *catB3*, encoding a chloramphenicol acetyltransferase, are found [288]. These findings indicate that integrons may participate in the acquisition of resistance to sulfamethoxazole, chloramphenicol and aminoglycosides in Bcc. 

The presence of plasmids in Bcc was studied early [289], but comprehensive information on their role in AR is still required. Something similar happens with bacteriophages. The finding of putative ARGs in prophages inserted in the chromosomes of different *B. cenocepacia* strains suggest that these genetic elements might be involved in the spread of resistance among Bcc [290]. However, detailed studies about the contribution of these elements in Bcc AR remain to be established.

## 6. Emerging Opportunistic Pathogens with Environmental Origin

Along with the opportunistic pathogens mentioned above, other environmental bacterial genera, such as *Brevundimonas, Shewanella, Achromobacter, Agrobacterium*, *Aeromonas*, *Erwinia* or *Pantoea*, among others, have been increasingly reported as responsible for emerging infectious diseases [291]. Note that while the most prevalent MDR opportunistic pathogens with a primary environmental habitat are non-fermentative Gram-negative bacteria, some environmental *Enterobacteriaceae* have been reported to cause human infections. Given their taxonomic relationship with highly prevalent human pathogens, as *E. coli* or *K. pneumoniae*, which easily acquire ARGs through HGT, the possibility that these environmental pathogens are a first step in the acquisition of ARGs by human bacterial pathogens [17,89,292,293] must be taken into consideration. 

*Brevundimonas* spp. are aerobic Gram-negative bacteria that are not only isolated from soils, submarine sediments and numerous aquatic habitats; but that also cause multiple types of infections, indicating that this genus may be a more widespread pathogen than previously thought [294]. *Brevundimonas diminuta* and *Brevundimonas vesicularis* have been isolated from clinical specimens, including blood, urine and lungs of CF patients [295,296,297,298,299]. The majority of *Brevundimonas* infections have been found in patients with underlying diseases, and many of them are acquired in hospitals [294,300]. Importantly, *Brevundimonas* infections are difficult to treat, as these bacteria can be resistant to different drugs, including fluoroquinolones or β-lactams [301,302]. Resistance to fluoroquinolones may be due to mutations in *gyrA*, *gyrB* and *parC* [301], and resistance to β-lactams to the presence of a VIM-2 MBL [302]. In addition, tetracycline resistance genes have also been detected in environmental isolates of *B. diminuta* [303]. Altogether, these data indicate that these bacteria should be considered as possible causes of nosocomial infections and should be included in prevention programs. Furthermore, their suggested use in bioremediation of contaminated seas and soils [304] should be carefully re-evaluated. 

Another microorganism with bioremediation potential is *Shewanella algae,* a marine bacterium [305] that also causes a variety of clinical symptoms in immunocompromised patients [306,307]. It has been suggested that some strains of *S. algae* isolated from clinical samples (skin ulcers and ear infections) [308,309,310,311] were mistakenly identified as *Shewanella putrefaciens* [312,313,314], a very close bacterial species [313]. A recent study has described the presence of β-lactams resistant clones of *S. algae* along the Italian Adriatic coast, containing AmpC and OXA-55-like β-lactamases [315]. Further, these authors have described the possible role of *S. algae* as a reservoir of ARGs, such as *qnrA* and β-lactamase genes (that confer resistance to quinolones and β-lactams, respectively), which could be transferred from the aquatic microbiota of Italian fish farms to bacteria of medical interest [316]. Actually, it has been proposed that *S. algae* are the origin of the quinolone resistance gene *qnrA*, widely distributed among plasmids present in several organisms [317], and different *Shewanella* species (as well as *A. baumannii*, see above) are considered as potential origins of some OXA-type β-lactamases [318]. Besides their contribution as progenitors of mobile ARGs, *Shewanella* can be involved in such mobility too. In fact, a plasmid harbouring several ARGs has been recently identified in *Shewanella xiamenensis* [319]. Although serious infections caused by *Shewanella* have been described [320,321], the rarity of these infections means that treatment guidelines have not been defined yet.

The genus *Achromobacter* is found in soils and aquatic environments, although some isolates can colonize the human intestinal tract, becoming opportunistic pathogens in immunosuppressed patients [322]. These bacteria can cause bacteraemia, meningitis and urinary tract infections [323,324,325]. Moreover, *Achromobacter* genus-belonging bacteria have also been isolated from CF patients [326,327,328]. *Achromobacter xylosoxidans* is the predominantly reported species among CF clinical isolates [322], but other *Achromobacter* species have also been isolated from these patients, such as *Achromobacter ruhlandiiand* [329]. They are intrinsically resistant to several drugs [328] due to the presence in their genomes of genes encoding RND MDR efflux pumps that extrude cephalosporins, aztreonam, carbapenems, quinolones, chloramphenicol, tetracyclines and erythromycin [330,331], as well as to the activity of β-lactamases [332,333]. Moreover, they are becoming increasingly resistant to carbapenems [328]. Furthermore, a recent study has described patient-to-patient transmission and AR development in different *Achromobacter* species [334]. Therefore, these species should be included in preventive programs.

The genus *Agrobacterium* is a recognized group of soil and plant-pathogenic bacteria that has also been implicated in human opportunistic infections, particularly *Agrobacterium radiobacter* (also known as *Rhizobium radiobacter*). Infections caused by these bacteria include bacteraemia, peritonitis and urinary tract infections, and they have been frequently associated with the use of intravascular devices in immunocompromised patients [335,336,337]. Further, *A. radiobacter* has been recently described to cause ocular infections, and it was identified in polymicrobial keratitis cases [338]. Although there is not much information about the intrinsic AR of these bacteria, the presence of RND efflux pumps in the genus *Agrobacterium* [339,340] suggests that these bacteria may present low susceptibility to different drugs. Further, the finding of an *A. radiobacter* clinical isolates carrying different antibiotic-inactivating enzymes [341] indicate that this microorganism may possess a wide set of AR determinants.

*Aeromonas* are Gram-negative bacteria with an aquatic environmental primary habitat that have also been suggested to behave as opportunistic pathogens [342]. Although there are controversial data about the role of these bacteria in human pathogenesis [343,344,345], different studies have described a significant correlation between diseases and the production of different virulence factors, such as haemolysins and enterotoxins [346,347]. The principal sources of these infections are contaminated water and foods, mainly inadequately cooked seafood and oysters [348,349]. In particular, *Aeromonas intestinalis*, *Aeromonas enterica*, *Aeromonas crassostreae* and *Aeromonas aquatilis* have been recently identified as representative species of *Aeromonas* with pathogenicity for both humans and aquatic organisms [350,351]. *Aeromonas* spp. are difficult to treat due to their intrinsic resistance to β-lactams, which results from a high constitutive expression of the gene encoding the β-lactamase AmpC, the low permeability of their external membrane and the activity of several outer membrane proteins [351,352]. In addition, *Aeromonas* species can also acquire resistance to β-lactams, such as ampicillin, and drugs from other structural families, such as erythromycin, tetracycline or chloramphenicol, by the acquisition of ARGs [353,354]. Moreover, these bacteria may have importance in aquatic environments as reservoirs of ARGs [355,356]. In this regard, it is worth mentioning that the analysis of the bacterial lineages likely associated with the dissemination of ARGs in a wastewater treatment plant indicate that *Aeromonas* could be a hub for such dissemination [11].

*Erwinia* is a genus of Enterobacteriales ubiquitous in the environment, especially in aquatic ecosystems and soils [357]. It mainly comprises phytopathogenic species, such as *Erwinia amylovora*, the first pathogen shown to cause disease in plants (i.e., the fire blight); *Erwinia persinicus*, which infects a wide range of hosts (e.g., tomatoes, cucumbers and bean pods); or *Erwinia carotovora*, among others. Strikingly, these phytopathogens and other plant-associated non-pathogenic *Erwinia* species (i.e., *Erwinia billingiae* and *Erwinia tasmaniensis*) have been occasionally found infecting animals, including humans [358]. For instance, *E. carotovora* and *E. persinicus* have exhibited pathogenicity against invertebrate infection models [357,359], and the latter has also been isolated from a human urinary tract infection [360]; whereas, *E. billingiae* and other non-phytopathogenic *Erwinia* strains can cause cutaneous infections, septic arthritis, brain abscesses or bacteraemia in humans [361,362,363,364]. Since these examples are quite unusual, the AR determinants that these species could harbour have not been sufficiently studied. However, there are reports about ARGs present in *E. amylovora*, like *strAB*, which codes for a phosphotransferase that confers resistance to streptomycin, and that has been likely acquired by non-pathogenic epiphytic bacteria also present in plant hosts [365]. In addition, some *E. amylovora* strains resistant to oxolinic acid, most likely mediated by chromosomal mutations, have been described [366]. Considering all this information, the potential of the *Erwinia* genus to become an opportunistic human pathogen, as other bacterial species described here, should be closely monitored. 

Another Gram-negative genus within the *Enterobacteriaceae* family is *Pantoea*, which includes 20 species isolated from different aquatic and terrestrial environments. Although many *Pantoea* isolates are misidentified, they have been described in association with plants and animals; mainly insects, but also birds, fish, bears, ruminants and importantly, humans [367,368,369]. The ability of this bacterial group to compete and survive in different niches has made it attractive for biotechnological uses. Water and soil isolates have been used for industrial applications, as bioremediation, since they are able to degrade many products; or agricultural purposes because they compete with plant pathogens and induce plant defences [370,371]. Besides being a plant pathogen, *Pantoea* has been recently identified in nosocomial environments. Different *Pantoea* species have been isolated from both immunocompetent and immunocompromised patients from wounds, blood, skin, stool, cysts and abscesses, as well as from urethra, trachea and oropharyngeal swabs [368]. Opportunistic infections in humans caused by *Pantoea* include septicaemia, pneumonia, septic arthritis, wound infections and meningitis [372]. As it happens with other pathogens with an environmental origin, clinical and environmental isolates are phylogenetically indistinguishable. Even more, *Pantoea* species considered primarily plant pathogens can be isolated from humans [367,369]. The most prevalent species infecting humans are *Pantoea agglomerans* and *Pantoea septica* [369]. Other clinically-relevant species include *Pantoea dispersa*, causing bacteraemia and neonatal sepsis [373], *Pantoea brenneri* and *Pantoea conspicua*, isolated from human sputum and blood, respectively [374,375]. Besides, clinical reports demonstrated cases of pneumonia and death in children with comorbidities where the causative agent was identified as an MDR *P. agglomerans,* resistant to third-generation cephalosporins, carbapenems, aminoglycosides and ciprofloxacin [376]. However, *Pantoea*’s AR determinants are mostly unexplored. Recently, a study has found that a foodborne *P. agglomerans* isolate possesses RND, ABC and MFS antibiotic efflux pumps such as MdtABC, MsbA or EmrAB, and antibiotic target modifiers that provide resistance to antibiotics such as macrolides, fluoroquinolones, tetracyclines or aminoglycosides [372]. Accordingly, further studies are needed to validate the ARGs of this opportunistic pathogen. 

Altogether, these data indicate that natural environments are an important primary source of opportunistic pathogens. Since humans, animals and natural environments are interconnected, One-Health approaches [8] are required to limit the spread and evolution of AR.

## 7. Ecological Role of Antibiotic Resistance Determinants Outside Clinical Settings

The environmental origin of different opportunistic bacteria indicates that the mechanisms of virulence and AR with a current role in human infection present a different and unique function in the natural environments where these bacteria emerge. Indeed, the fact that intrinsic ARGs may have other functional roles besides AR has been previously discussed [17,377,378,379,380]. While some of these functions deal with basic aspects of bacterial physiology, such as peptidoglycan recycling [381], some others are related to bacterial interactions with other elements of the biosphere and hence, have ecological value. This includes not only ARGs but also situations that trigger transient AR. For instance, it has been demonstrated that increased production of alginate, a key element for *P. aeruginosa* biofilm formation, protects this bacterial species against its protozoan predators in nature [382]. Concerning ARGs, bacterial MDR efflux pumps stand as relevant elements modulating bacterial interactions with the environment. These ARGs are ancient elements that extrude not only antibiotics but also a wide range of non-antibiotic substrates. Further, the facts that efflux pumps are conserved within a species and between species [383,384], that their expression may be induced by host-produced compounds [246,385,386,387,388,389,390,391,392] (such as bile salts or fatty acids, plant-produced compounds or QS signals, from humans, plants and bacteria, respectively), and that these systems are able to extrude non-antibiotic substrates [393] (such as QS signals, bacterial metabolites, or plant-produced compounds [390,392,394,395,396,397,398,399]), indicates that they play important roles in the adaptation of bacterial physiology to changing environments. In this review, we discuss the role of efflux pumps outside clinical settings, focusing on bacterial interactions in the rhizosphere.

The rhizosphere is a natural ecosystem that comprises the plant roots and microbial community present in the surrounding soil. Within this ecosystem, soil bacteria and plants affect each other, leading to a feedback system that drives the ecology and evolution of both organisms [400]. Accordingly, the evolution of the microbial community is the result of either the trade-offs associated with overcoming the plants’ defence or the specific benefits associated with the host plant colonization. In this sense, plants’ roots, apart from providing mechanical support and allowing the absorption of water and nutrients by plants, exudate a wide array of natural products into the rhizosphere [401]. This extrusion modifies soil composition and provides both nutrients for bacterial growth and defensive secondary metabolites. Therefore, roots shape the composition and dynamics of microbial communities, as only bacteria capable of dealing with root exudates are present in the rhizosphere, but they also drive the evolution of plant pathogens [402]. The selection of more virulent mutants that can evade plant defences [403,404] and of mutants that present an improved capacity to metabolize plant-produced nutrients is the driving force of this evolution [405]. Even more, microorganisms present different mechanisms that allow them to deal with root exudates, such as the flavonoid-responsive family of RND efflux pumps. These mechanisms of resistance have been identified in different plant-associated bacteria such as *Agrobacterium tumefaciens* [340], *Pseudomonas syringae* [390,406], *E. amylovora* [392,407], *Bradyrhizobium japonicum* [408], *Xanthomonas axonopodis*, *Ralstonia solanacearum* [409], *S. maltophilia* [246] and *Sinorhizobium meliloti* [410].

As mentioned above, MexAB-OprM is an important MDR determinant of the human opportunistic pathogen *P. aeruginosa* [411,412], which contributes to its intrinsic resistance to several antibiotics (quinolones, macrolides, tetracycline, chloramphenicol and β-lactams) [413]. Besides, this pump is a relevant mechanism for acquiring AR in clinical settings [414] since *mexAB-oprM* overexpressing mutants are selected in the infected patients [415]. MexAB-OprM is also able to extrude monoterpenes and related alcohols present in the tea tree (*Melaleuca alternifolia*) [394], indicating a role in natural environments that was probably acquired before that of antibiotic resistance at clinical settings. In fact, plant flavonoids induce the expression of *mexAB-oprM* in *P. syringae*, the causal agent of bacterial speck in tomato plants, which allows colonization of these plants [390]. These compounds are inhibitors of motility and the type III secretion system in *P. syringae* via the GacS/GacA two-component system [416,417]. Therefore, one of the roles of MexAB-OprM in natural environments is extruding flavonoids to avoid the inhibition of virulence and hence, allowing the colonization of tomato plants. In fact, flavonoids also regulate the capacity of other plant-associated bacteria to colonize plants, such as *E. amylovora*, *A. tumefaciens, X. axonopodis* and *S. maltophilia*. These effectors are inducers and may also be substrates of efflux pumps [246,407,418,419]. These Red-Queen adaptive coevolution phenomena indicate that the original role of bacterial efflux pumps may be the extrusion of plant-derived anti-virulence compounds, among others. Therefore, the screening of natural or natural-like compounds that act as both inducers and substrates of efflux pumps of clinical relevance could serve to identify virulence inhibitors that could be potentially combined with antibiotics in new therapeutic strategies to control bacterial infections caused by environmental pathogens as *P. aeruginosa* [420].

Root exudates not only avoid colonization by pathogenic bacteria but also recruit nitrogen-fixing and growth-promoting bacteria [421]. Many plant species, mainly legumes, present an intimate association with nitrogen-fixing bacteria and, again, the above-mentioned flavonoids are involved in establishing these associations [422]. This has been observed in *S. meliloti* [410,423] and *B. japonicum* [408], in which flavonoids are also inducers of efflux pumps. Additionally, roots also attract bacteria able to promote plant growth by the extrusion of carbohydrates, amino acids and benzoxazinoids [424,425]. Once again, efflux pumps may be mediating these associations.

Efflux pumps also play essential roles in bacteria-bacteria interactions within the host plant, where there is competition for space and nutrients. Cell-cell interactions are controlled by the QS system, which allows cooperation within a species to colonize a given environment and inter-species communication. In this regard, it is known that the AR determinants MexAB-OprM and MexCD-OprJ of *P. aeruginosa* [411,412,426] modulate QS-responses and host-pathogen interactions, either by the extrusion [427,428,429,430] or by the impaired production [431] of QS signals or their metabolic precursors. While bacteria from the rhizosphere produce QS signals to coordinate plant colonization [432], plants may secrete compounds similar to bacterial N-acyl-homoserine lactones (AHLs) through root exudation [433,434], something that is known as Quorum Quenching. For instance, the red seaweed *Delisea pulchra* produces halogenated furanones that interfere with the AHL regulatory system of several Gram-negative bacteria [435,436]. In addition, it is known that certain bacteria also possess the ability to quench QS by enzymatic degradation of AHL signals [437]. This is the case of a *Bacillus* acyl-homoserine lactonase enzyme able to hydrolyse the lactone bond of AHL compounds of the plant pathogen *E. carotovora* [438]. 

Finally, another relevant role of efflux pumps in bacteria-bacteria interactions within the plant host is the extrusion of antimicrobial compounds produced by other bacterial species. For example, *E. amylovora* and *P. agglomerans* (a biocontrol agent for fire blight) co-colonize rosaceous plants [439,440], but the last one impedes colonization of stigmas of apple and pear plants by *E. amylovora* by effectively inhibiting its growth [441]. However, this microorganism can reach high-density populations when the expression of *norM* is induced (at 18 °C [439]), indicating that this efflux pump extrudes antimicrobial compounds produced by *P. agglomerans* [396]. 

All in all, these data indicate that bacterial efflux pumps are much more than AR determinants. They are relevant elements for the physiology of microorganisms in natural ecosystems. In this sense, it is important to keep in mind that evolution is similar to a tinkerer [442], which produces new functions from old materials and not from scratch.

**Table 1 ijms-22-08080-t001:** Clinically relevant MDR efflux pumps in *P. aeruginosa*

Efflux Pump	Main Regulators	Substrate Range	Resistance	References
MexAB-OprM	MexR, NalD, NalC	β-lactams (excepting imipenem), quinolones, macrolides, tetracyclines, chloramphenicol	IR *, AR **, TR ***	[384]
MexCD-OprJ	NfxB	Penicillin, cefepime, cefpirome, meropenem, quinolones, macrolides, tetracyclines, chloramphenicol	AR, TR	[413]
MexEF-OprN	MexT, MexS	Carbapenems, quinolones, chloramphenicol	AR, TR	[443]
MexXY-OprM	MexZ	Penicillin, cefepime, cefpirome, meropenem, quinolones, macrolides, tetracyclines, chloramphenicol, aminoglycosides	IR, AR, TR	[413]

* Intrinsic (IR) ** Acquired (AR) and *** Transient (TR) antibiotic resistance.

**Table 2 ijms-22-08080-t002:** Main antibiotic resistance determinants encoded in *S. maltophilia* genome.

Gene	Product	Drug Resistance	Type of Resistance	References
*L1*	Class B3 Zn^2+^-dependent MBL	β-lactams (except monobactams)	IR *, TR ***	[208,211]
*L2*	Class A clavulanic acid-sensitive cephalosporinase	β-lactams	IR, TR	[209,211]
*aac(6′)-Iz*	N-Aminoglycoside acetyltransferase	Amikacin, tobramycin, sisomicin, netilmicin	IR	[214]
*aac(6′)-Iak*	N-Aminoglycoside acetyltransferase	Arbekacin, kanamycin, neomycin, sisomicin, tobramycin	IR	[213]
*aph(3′)-IIc*	Aminoglycoside phosphotransferase	Kanamycin, neomycin, paromycin, butirosin	IR	[215]
Sm*qnr*	Pentapeptide repeat protein	Quinolones	IR, AR **	[218,219,240]
*smeYZ*	RND efflux pump	Aminoglycosides, tetracycline, leucomycin, SXT	IR, AR, TR	[247,444]
*smeDEF*	RND efflux pump	Fluoroquinolones, chloramphenicol tetracycline, tigecycline, macrolides, sulfamethoxazole, trimethoprim, SXT	IR, AR, TR	[207,225,229,445,446]
*smeGH*	RND efflux pump	β-lactams, fluoroquinolones, tetracycline, polymyxin B, ceftazidime	IR, AR	[221]
*smeIJK*	RND efflux pump	Aminoglycosides, tetracycline, ciprofloxacin, levofloxacin, leucomycin, minocycline	IR, AR	[222]
*smeOP*	RND efflux pump	Nalidixic acid, doxycycline, aminoglycosides, macrolides	IR	[223]
*smeVWX*	RND efflux pump	Quinolones, chloramphenicol, trimethoprim/sulfamethoxazole	AR, TR	[225,226,247,248]
*smeABC*	RND efflux pump	Aminoglycosides, β-lactams and fluoroquinolones	AR	[238]
*smrA*	ABC efflux pump	Fluoroquinolones, tetracycline, doxorubicin	ND	[230]
*macABCsm*	ABC efflux pump	Aminoglycosides, macrolides, polymyxins	IR	[231]
*emrCABsm*	MFS efflux pump	Nalidixic acid, erythromycin, CCCP, tetrachlorosalicylanilide	IR	[232]
*fuaABC*	Fusaric acid tripartite efflux pump	Fusaric acid	TR	[249]

* Intrinsic (IR) ** Acquired (AR) and *** Transient (TR) antibiotic resistance.

## Figures and Tables

**Figure 1 ijms-22-08080-f001:**
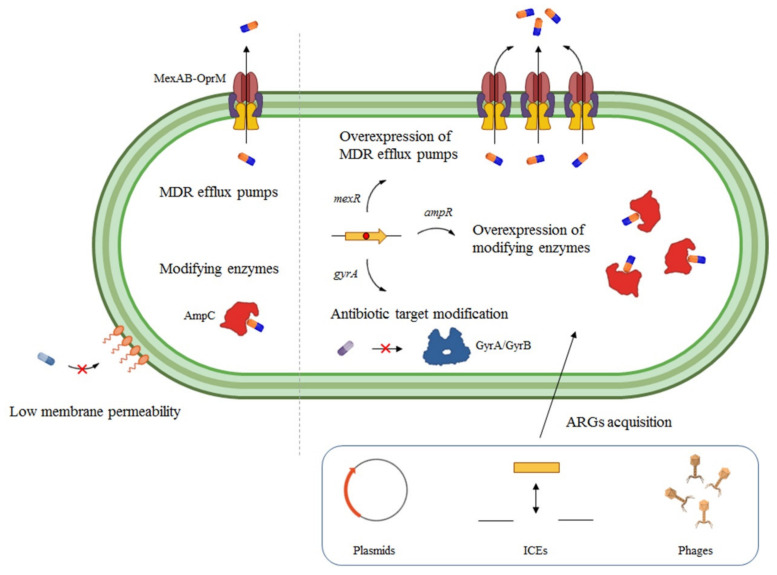
Schematic representation of the main elements involved in intrinsic and acquired antibiotic resistance in *Pseudomonas aeruginosa*. *P. aeruginosa* possesses a remarkable intrinsic resistance to antibiotics caused, among other factors, by the production of antibiotic-modifying enzymes (e.g., β-lactamase AmpC), low outer membrane permeability and a great amount of multidrug resistance (MDR) efflux pumps like MexAB-OprM. Antibiotic resistance level may increase by chromosomal mutations in genes encoding negative regulators of the above-described intrinsic resistance determinants, such as genetic modifications within *mexR* or *ampR,* which boost the expression of *mexAB-oprM* and *ampC*, respectively. The modification of the antibiotic target is also a frequent mechanism for acquiring antibiotic resistance in *P. aeruginosa*, as the increased resistance to quinolones by mutations in gyrases encoded by *gyrA* or *gyrB*. Alternatively, this bacterium is also able to acquire novel ARGs, which are located in mobile elements, such as plasmids or integrative and conjugative elements (ICEs). It has been stated that bacteriophages might also be involved in the acquisition of ARGs, but the role of these genetic elements in the spread of resistance in *P. aeruginosa* is not yet fully understood.

## Data Availability

Not applicable.

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
