# Peer review of "Coming from the Wild: Multidrug Resistant Opportunistic Pathogens Presenting a Primary, Not Human-Linked, Environmental Habitat"

_ijms, 2021, doi:10.3390/ijms22158080_

Round 1

Reviewer 1 Report

In this revised manuscript by Sanz-Garcia, the authors presented an review on antibiotic resistant bacterial pathogens with environmental origins. Overall, the authors presented detailed review of the various virulence mechanism of certain well-characterized opportunistic pathogens as well as some novel/less studied environmental bacteria with the potential to become human pathogen.  The writing are good but I wish the authors could make figures illustrating the resistance mechanism of these pathogens. A picture is worth a thousand words and I would recommend the authors to include a few.

Minor comments

  • Bacterial names not in italic when it's shown as headings
  • Text in Table 2 should be in single space

Author Response

We appreciate the positive opinion of the referee concerning our work.

In agreement with the suggestions, a new figure has been included and the minor comments have been addressed.

Reviewer 2 Report

The authors presented a comprehensive review on MDR opportunistic pathogens with an environmental origin, including a complete explanation of the antibiotic resistance profiles, mechanisms of resistance, and population structure of these species. One of the most interesting discussed aspects of the article is the original play of many ARGs, such as the bacterial MDR efflux pumps with primary roles in environmental lifestyle, which falls in the One-health perspective on ARG and MDR dynamics. The paper is very well-written and the information precise and updated, then I only have few minor comments to improve the manuscript.

The authors aim to discuss how genes conferring resistance to antibiotics generally have another functional role in the environmental life of the carriers. This aspect in the text could be a little more extended, including other examples beyond the MDR efflux pumps since this dual role (clinical-ecological) is quite relevant.

For the pathogens characterized, more information on the role of bacteriophages in antibiotic resistance profiles and transmission and possible environmental play would be valuable.

Regarding this quote, “It is important noticing that, besides being the primary habitat and thus the origin of some opportunistic pathogens nowadays causing problems at hospitals, natural ecosystems are the sink where all human bacteria, pathogens, and commensals, together with the ARGs they carry, end up,” the authors should clarify if they refer to the wastewater, the circulating water, or the sinks.

In this sentence, “Overall, these features result in P. aeruginosa being subsumed into two bacterial ensembles (ESKAPE and TOTEM), which include the currently most relevant multidrug resistant human pathogens,” indicate what acronyms ESKAPE and TOTEM mean.

Author Response

We appreciate the positive opinion of the referee concerning the article.

In agreement with the suggestions some more information on bacteriophages and on the role or ARGs beyond antibiotic resistance has been included. 

Concerning the quote of sink, we refer here to the overall non-clinical environment as a sink, with our further details. The sentence ha been modified for clarification.

The meaning of ESKAPE and TOTEM has been clarified.